# Shotgun metagenome sequencing identification of a set of genes encoded by *Actinomyces* associated with medication-related osteonecrosis of the jaw

Hiroko Yahara[1] *, Akimitsu Hiraki[2], Yutaka Maruoka[3], Aki Hirabayashi[4], Masato Suzuki[4], Koji Yahara[4] *

1 Genome Medical Science Project (Toyama), Research Institute, National Center for Global Health and Medicine, Tokyo, Japan, 2 Section of Oral Oncology, Department of Oral and Maxillofacial Surgery, Fukuoka Dental College, Fukuoka, Japan, 3 Department of Oral and Maxillofacial Surgery, Center Hospital, National Center for Global Health and Medicine (NCGM), Tokyo, Japan, 4 Antimicrobial Resistance Research Center, National Institute of Infectious Diseases, Higashimurayama, Tokyo, Japan

☯ These authors contributed equally to this work.
* h-yahara@ri.ncgm.go.jp (HY); k-yahara@nih.go.jp (KY)

**Data Availability Statement:** The sequence data after the preprocessing described below and removal of human reads of the healthy control

## Abstract

Medication-related osteonecrosis of the jaw (MRONJ) is intractable and severely affects a patient's quality of life. Although many cases of MRONJ have been reported in the past decade, the disease pathophysiology is unclear and there are no evidence-based therapeutic strategies. MRONJ usually features bone inflammation and infection. Prior studies that explored the association between MRONJ and microbial infection used the culture-based approach, which is not applicable to hundreds of unculturable taxa in the human oral microbiome, or 16S ribosomal RNA gene sequencing, which does not provide quantitative information of the abundance of specific taxa, and information of the presence, abundance, and function of specific genes in the microbiome. Here, deep shotgun metagenome sequencing (>10 Gb per sample) of bulk DNA extracted from saliva of MRONJ patients and healthy controls was performed to overcome these limitations. Comparative quantitative analyses of taxonomic and functional composition of these deep metagenomes (initially of 5 patients and 5 healthy controls) revealed an average 10.1% increase of genus *Actinomyces* and a 33.2% decrease in genus *Streptococcus* normally predominant in the human oral microbiota. Pan-genome analysis identified genes present exclusively in the MRONJ samples. Further analysis of the reads mapping to the genes in the extended dataset comprising five additional MRONJ samples and publicly available dataset of nine healthy controls resulted in the identification of 31 genes significantly associated with MRONJ. All these genes were encoded by *Actinomyces* genomic regions. Of these, the top two abundant genes were almost exclusively encoded by *Actinomyces* among usual taxa in the human oral microbiota. The potential relationships of these key genes with the disease are discussed at molecular level based on the literature. Although the sample size was small, this study will aid future studies to verify the data and characterize these genes in vitro and in vivo to understand the

(BioProject PRJDB10432) and those of the patients (BioProject PRJDB10606) are publicly available.

**Funding:** This research was supported by a Grant-in-Aid for Scientific Research of Education, Culture, Science, Sports, and Technology (MEXT) of Japan (19H04846 to K.Y., 19J40070 to H.Y. and 16H06429, 16K21723), and in part by Grants-in-Aid for Research from the National Center for Global Health and Medicine (20A-3002 to H.Y.). The funders had no role in study design, data collection and analysis, decision to publish, or preparation of the manuscript.

**Competing interests:** The authors have declared that no competing interests exist.

disease mechanisms, develop molecular targeted drugs, and for early stage screening and prognosis prediction.

## Introduction

Bisphosphonates (BPs) reduce bone resorption by inhibiting osteoclasts. BPs are widely used to treat many diseases, including malignant neoplasm, multiple myeloma, and osteoporosis [1]. Patients treated with BPs and invasive dental treatment sometimes suffer BP-related osteonecrosis of the jaw (BRONJ), which has been reported since 2003 [2]. BRONJ has also been reported among patients treated using the denosumab human monoclonal antibody that binds to the receptor activator of nuclear factor-kappa B ligand (RANKL). Denosumab treatment can reduce bone resorption, which prompted the general term "antiresorptive agent-induced ONJ" (ARONJ) [3] that corresponds to the disease related to the administration of antiresorptive therapeutic agents. The American Association of Oral and Maxillofacial Surgeons (AAOMS) proposed the general term of medication-related osteonecrosis of the jaw (MRONJ) in a 2014 position paper [4], since the condition can also occur in patients treated with biological medications that lack antiresorptive properties [5]. MRONJ is intractable and severely affects a patient's quality of life by causing pain and difficulty in eating and worsening oral hygiene. Although many cases of MRONJ have been reported in the past decade, the disease pathophysiology is unclear and there are no strictly-defined therapeutic strategies [6]. MRONJ remains a major concern with dentists, oncologists, rheumatologists, and general practitioners [7].

According to a consensus report published by the International Task Force on Osteonecrosis of the Jaw, the incidence of BRONJ in osteoporosis patients treated with low-dose oral BPs ranges from 1.04 to 69.0 per 100,000 patient-years, while that in bone cancer patients treated with high-dose intravenous BPs ranges from 0 to 12,222 per 100,000 patient-years [8]. The Japanese Society of Oral and Maxillofacial Surgeons identified 4,797 cases from January 2011 through December 2013, in which patients who received oral BPs accounted for 49.2% [9]. A previous study reported 67% of BRONJ cases were preceded by tooth extraction, which likely triggered onset [10]. Another study reported tooth extraction was associated with a 18 times higher incidence of BRONJ [11].

BPs have been the most popular antiresorptive drugs. BPs act by binding the mineral component of bone and interfering with the action of osteoclasts. The discovery of RANKL and the essential role of RANK signaling in osteoclasts led to the development of denosumab. Although BPs and denosumab are mechanistically distinct antiresorptive agents, the incidence of MRONJ is similar [12]. A characteristic of MRONJ is its association with the use of the antiresorptive therapies. Thus, quantification of bone resorption was previously explored for prognosis. No marker was established [8]. No single model can fully explain morphological changes observed in MRONJ at the macro- and microscopic levels. However, several unique features are evident. MRONJ lesions are characterized by scattered areas of necrotic bone, as evidenced by empty osteocyte lacunae [13]. Secondly, bone inflammation and infection are usually associated with MRONJ, and bone invasion by *Actinomyces* was observed in 82.18% of patients [14].

Previous studies that explored associations between MRONJ and infection of microbes were based on culturing, mass spectrometry, or sequencing of the 16S ribosomal RNA (rRNA) gene [15, 16]. Given that over nearly 250 of approximately 700 taxa at the species level in the human oral bacteriome are as yet uncultivated [17], the culture-based approach has obvious

limitations and might have missed unculturable microbes associated with MRONJ. While 16S rRNA gene sequencing improved the limitations, it cannot give quantitative information of abundance of specific taxa in the microbiome. Therefore, the contribution to MRONJ of the increased prevalence of specific microbial taxa in the oral microbiome remains unknown. Furthermore, 16S rRNA gene sequencing provides no information of the presence, abundance, and function of specific genes in the microbiome.

Here, we conducted shotgun metagenome sequencing (>10 Gb per sample) of bulk DNA extracted from saliva of MRONJ patients and healthy controls collected and stored using a kit specialized for microbial and viral DNA/RNA. The approach overcame the limitations of 16S rRNA gene sequencing. Through comparative quantitative analyses of taxonomic and functional composition of these deep metagenomes, we identified a set of microbial genes almost specific to MRONJ that are exclusively encoded by *Actinomyces* and increased in abundance.

## Materials and methods

### Diagnostic criteria

This study followed the AAOMS definition of BRONJ, which requires the presence of all of the following: 1) current or previous treatment with BP; 2) no history of radiation therapy to the jaws or obvious metastatic disease to the jaws; and 3) exposed bone or bone that can be probed though an intra or extraoral fistula in the maxillofacial region that has persisted > 8 weeks after identification by medical staff [4]. To exclude the effect of antibiotic treatment, we included only cases before treatment for MRONJ (but not the underlying condition). The inclusion criteria for this were 0 and 1 stage BRONJ based on a previous position paper [18].

### Sample collection, DNA extraction, and metagenome sequencing

Based on previous reports that microbial profiles within subjects were stable throughout a 24-h period [19] and the similar profiles of unstimulated and stimulated saliva [20], 1 mL unstimulated saliva was collected and stored using a kit specialized for microbial and viral DNA/RNA (OMNIgene ORAL OM-501) from 5 MRONJ patients (one man, 4 women; 56-to-95 years old). The participants were instructed not to eat, drink, brush their teeth, or gargle for at least 1 h prior to sampling. Unstimulated saliva had recently been collected from 4 healthy volunteers and stored in the same way. The samples were used for DNA extraction and metagenome sequencing as described below. The metagenome data are publicly available (accession DRR214959- DRR2149562 in NCBI BioProject PRJDB9452) and were downloaded and used in the present study. Therefore, 1 mL of unstimulated saliva was additionally collected and stored from a healthy volunteer. The healthy volunteers were 2 men and 3 women aged 35-to-65 years.

The saliva samples stored in the kit were subjected to DNA extraction using an enzymatic method [21]. The extracted DNA samples were stored in 50 μL pure water and used for Nextera XT library construction. The libraries were mixed and subjected to two multiplex genome sequencing runs with the Illumina HiSeq $2 \times 150$ bp paired-end run protocol. The amount of raw sequence data was 49.3, 128.6, 14.8, 16.4, 34.2 Gb for the 5 MRONJ patients, and 11.7 Gb for the healthy control. The publicly available metagenome data are > 30 Gb for each of the 4 healthy controls.

The sequence data after the preprocessing described below and removal of human reads of the healthy control (BioProject PRJDB10432) and those of the patients (BioProject PRJDB10606) are publicly available.

## Preprocessing, taxonomic profiling, assembly, and gene-by-gene analyses

EDGE pipeline version 1.5 [22] was used for preprocessing (trimming or filtering out reads, and removal of reads mapped to the human genome) of the HiSeq data. Reads of 0.15–2.51% reads were discarded, and 0.70–4.86% of bases were trimmed in the initial quality control. Then, 0.06–2.78% of the filtered reads that mapped to the human genome were removed. Taxonomic profiling was conducted using Kraken [23] and its full database (created in March 2017 according to https://github.com/mw55309/Kraken_db_install_scripts) implemented in metaWRAP [24] as a module can directly subsample the same number (approximately 37 million) of read pairs in each sample. The preprocessing of the sequencing data and taxonomic profiling were also performed for the five additional MRONJ samples in an extended dataset explained in the next section.

HiSeq reads were assembled using SPAdes [25] with the "—meta" option. We then predicted the protein-coding genes for every contig using Prokka [26], and constructed a gene presence or absence matrix for the entire set of genes (i.e., orthologous clusters) detected among the case and control samples using the Roary pipeline [27] with the "-i 90—group_limit 1000000" option. For each gene found to be specific to MRONJ samples, gene function annotation was performed by searching the database of hidden Markov models profiles of orthologous genes defined in the KEGG database using KofamScan [28].

## Mapping reads to the extended dataset and estimation of their abundance

The MRONJ sample-specific genes were further examined in an extended dataset of five additional MRONJ patients (5 females; aged 44–87 years) and publicly available salivary metagenome data of nine healthy controls (all seven datasets of the Human Microbiome Project (https://www.hmpdacc.org/HMIWGS/healthy) along with two Japanese datasets (accession numbers: DRR046069 and DRR046087) [29]). Saliva collection, DNA extraction, and library construction were conducted for the five additional MRONJ patients using the same protocol as above. The libraries were separately subjected to genome sequencing with the Illumina HiSeq 2 × 150 bp paired-end run protocol. The size of the raw sequence data was 35.8, 32.0, 31.2, 31.2, 33.1 Gb for the five additional MRONJ samples. The sequencing reads of each sample were mapped to the nucleotide sequences of the genes identified above using KMA ($k$-mer alignment) [30]. The nucleotide sequences of the genes of the 1st MRONJ patient were used as templates. The presence of genes was determined based on the following criteria: > 90% nucleotide sequence identity between the reads and the template gene sequence, and reads covering > 80% of the length of the template sequence.

Aligned nucleotide sequences of the genes exhibiting > 80% increase in frequency in MRONJ samples compared to their frequency in the healthy controls in the extended dataset were extracted from the KMA output file. The relative abundance of each gene in each sample was calculated as transcripts (reads) per million (TPM) [31] using CoverM (https://github.com/wwood/CoverM), based on the mapping of HiSeq reads with the extracted nucleotide sequence of each MRONJ sample. The TPM values could be directly compared among the samples because the sum of the values was normalized to 1 million [31].

## Taxonomic analysis of contigs and genes associated with MRONJ

The Contig Annotation Tool (CAT) [32] was applied for taxonomic classification of the contigs based on amino acid sequence searching of each ORF against the NCBI nr database followed by a voting approach by summing all scores from ORFs separately supporting a certain taxonomic classification (superkingdom, phylum, class, order, family, genus, and species) and checking if the summation exceeded a cutoff value (by default, 0.5 × summed scores)

supporting a superkingdom of the certain taxonomic classification across ORFs, representing a balance between the classification precision and a fraction of the classified sequences.

The presence or absence of each gene associated with MRONJ and encoded in a contig assigned to *Actinomyces* was checked among publicly available genome sequences of *Actinomyces* spp. strains. We also constructed a core genome alignment of the *Actinomyces* species using the Roary pipeline with the "-i 80" option accounting for high genetic diversity among the species. We then constructed a maximum-likelihood tree using PhyML [33] from the alignment. The presence of MRONJ-associated genes for each strain were defined according to the criterion of >90% nucleotide sequence identity over > 80% of the locus length, and were illustrated as a heatmap using Phandango [34].

Phylogenetic distribution of the MRONJ-associated genes outside *Actinomyces* was examined using BLASTN searches against the KEGG GENOME database. The results were tabulated using an in-house Perl script with a cutoff of e-value 1e-90 indicative of gene presence. For genera including at least an MRONJ-associated gene, we extracted nucleotide sequences of 16S rRNA from representative strains registered in the Microbial Genome Database for Comparative Analysis (MBGD) or NCBI Genbank. The nucleotide sequences were aligned using MAFFT. Similarly, a maximum-likelihood tree was constructed using PhyML from the alignment. The presence or absence of the MRONJ-associated genes for each genus were illustrated as a heatmap using Phandango.

### Ethical considerations

This study was approved by the ethics committees of the Research Institute National Center for Global Health and Medicine (approval number NCGM-G-0002529-02), National Institute of Infectious Diseases (811), Fukuoka Dental University (368). Written informed consent was obtained from the participants of the study.

## Results

### Taxonomic profiling identifies microbes associated with MRONJ

Taxonomic profiling (i.e., computational inference of taxonomic clades populating a given microbial community and their proportions; relative abundance) using the deep shotgun metagenome data (>10 Gb per sample) revealed an average 24.8% increase in *Actinobacteria* phylum among the five MRONJ patients compared to the five healthy controls (Fig 1). A 33.2% decrease in *Streptococcus* genus normally predominant in the human oral microbiota was evident in MRONJ patients (Fig 1). Among *Actinobacteria*, genus *Actinomyces* increased 10.1% on average. No other genus displayed >10% increased/decreased abundance in MRONJ patients. The average relative abundance of *Actinomyces* among the five MRONJ patients was 14.0%, which was almost the same (14.4%) among five more MRONJ patients in the extended dataset used in the next section.

### MRONJ-associated genes encoded in *Actinomyces* genomic regions

A gene presence or absence matrix was created by assembling the metagenomic reads, and gene finding and analysis of the pan-genome (i.e., the entire repertoire of genes encoded in the assembled contig sequences) among the five MRONJ patients compared to the five healthy controls, which revealed 191 genes present in all MRONJ samples and absent in any healthy control sample. Further analysis of the reads mapped to the genes in the extended dataset comprising five additional MRONJ samples and the publicly available salivary metagenome dataset of nine healthy controls resulted in the identification of 31 genes, which were present in 90%

A

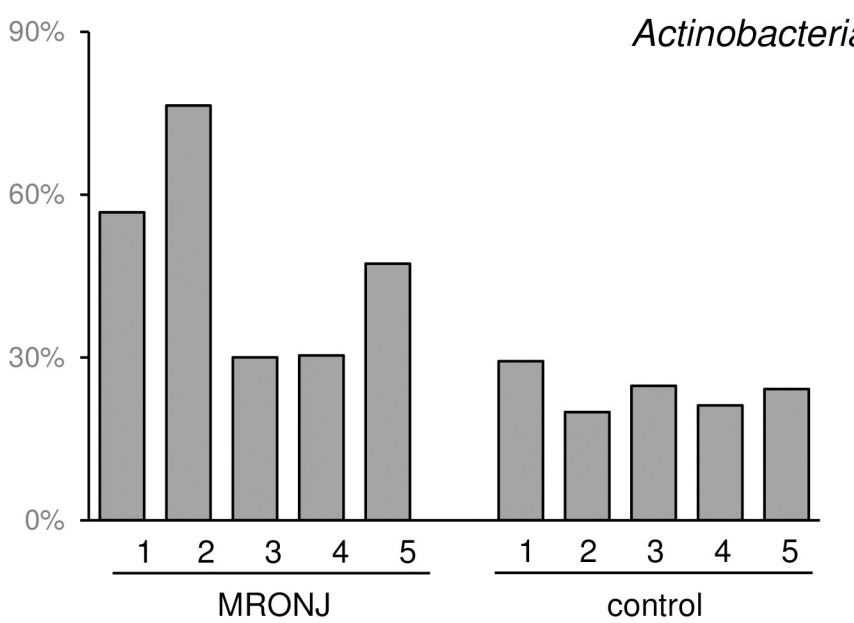

B

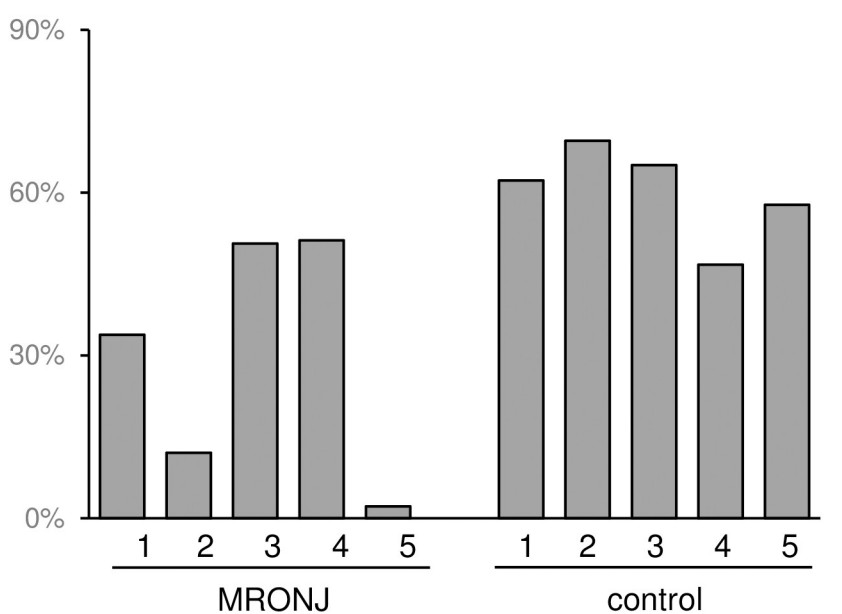

**Fig 1. Relative abundance of MRONJ-associated microbes in each MRONJ case and healthy control.** (A) *Actinobacteria* and (B) *Streptococcus*.

(A)

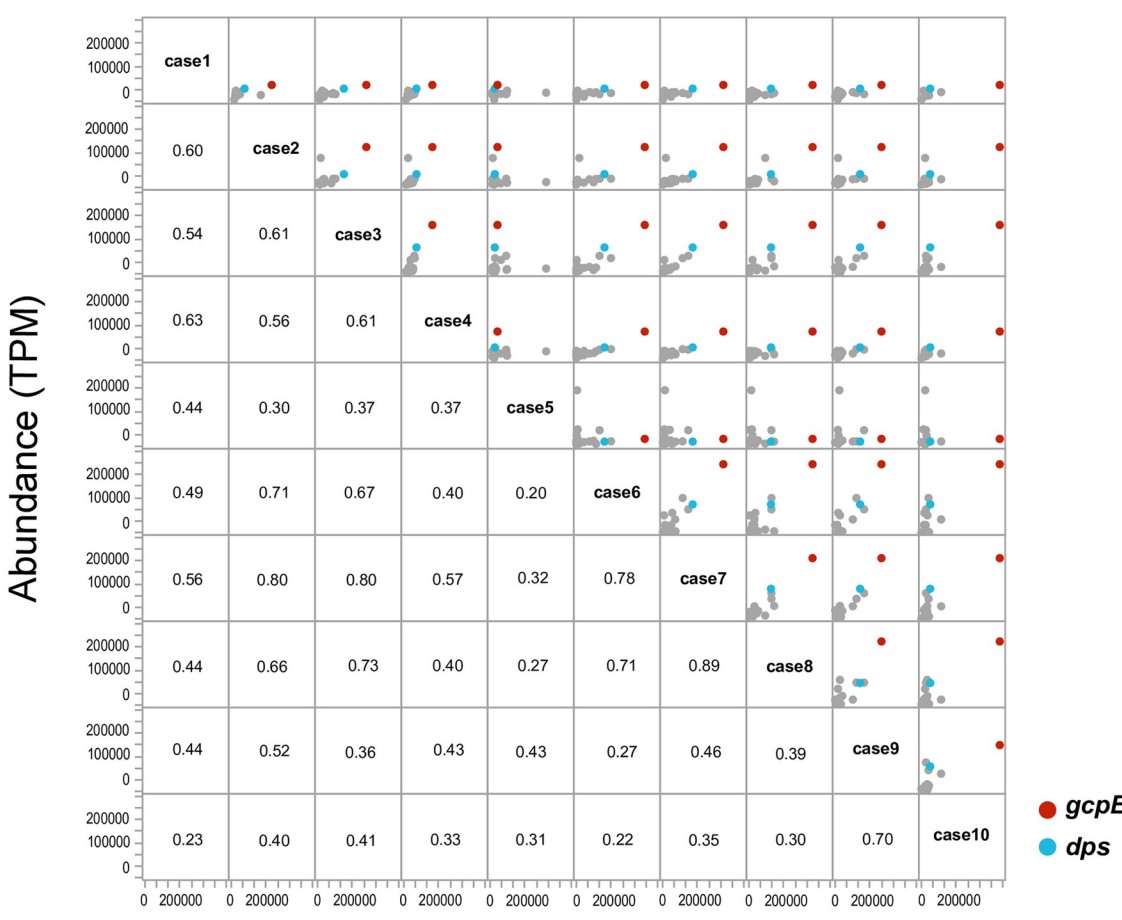

Abundance (TPM)

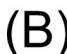

Abundance (TPM)

(B)

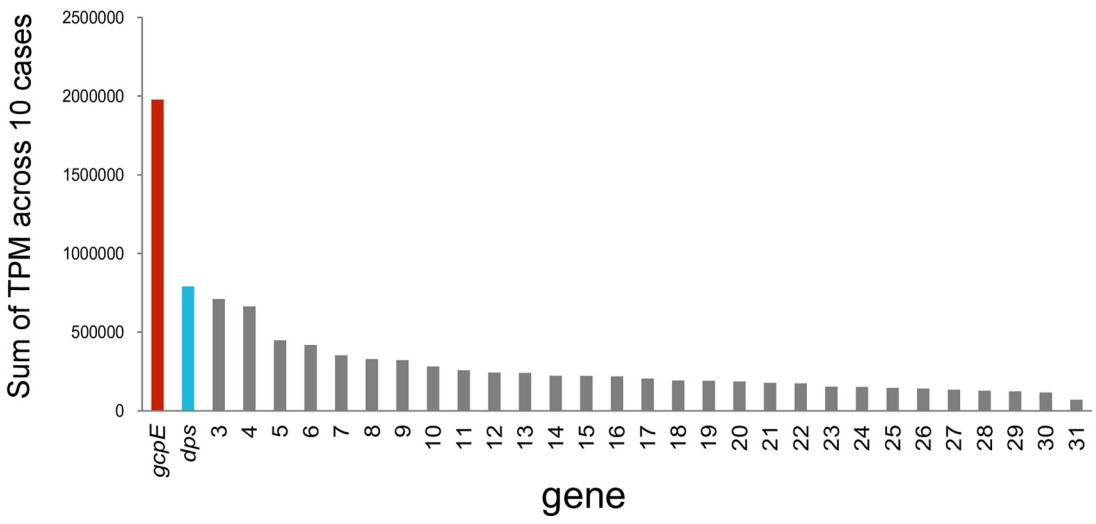

**Fig 2. Abundance of MRONJ-associated genes in ten MRONJ patients.** (A) Pairwise comparison. Each dot corresponds to a gene. The values indicate Spearman's correlation coefficient calculated for each patient pair. The colored dots indicate the two most abundant genes shown in (B) and described in the main text (red; *gcpE*, and blue; *dps*). (B) MRONJ-associated genes were ranked based on the sum of TPM values.

of the MRONJ samples (i.e., present in 9 out of the 5 plus 5 samples) and 0 or 7% of the healthy control samples (i.e., present in 0 or 1 sample out of the 5 plus 9 samples) ($p < 0.005$ after Bonferroni correction, Fisher's exact test). Nucleotide sequences of the 31 genes are available at https://figshare.com/articles/dataset/Nucleotide_sequence_of_MRONJ-associated_31_genes/13023143.

Taxonomic classification of assembled contigs encoding the 31 genes based on an amino acid sequence search of each open reading frame (ORF) against the NCBI nr database followed by summation of all scores from the ORFs revealed that they were all assigned to genus *Actinomyces*.

The abundance of each of the 31 genes (S1 Table) plotted for each pair of the 10 MRONJ patients is shown in Fig 2A. The Spearman's correlation coefficient between gene abundance was higher than 0.3 among 84% of the paired MRONJ samples. When the 31 genes were ranked based on the sum of TPM values in 10 patients, the top two genes were *gcpE* (encoding (E)-4-hydroxy-3-methylbut-2-enyl-diphosphate synthase) and *dps* (encoding starvation-inducible DNA-binding protein) (Fig 2B).

## Phylogenetic distribution of MRONJ-associated genes

A core genome phylogenetic tree in the *Actinomyces* genus was constructed for 51 strains registered in the public databases (S2 Table), together with information of the presence or absence of the 31 MRONJ-associated genes in each strain (Fig 3). Concerning the distribution of the 31 MRONJ-associated genes, strains harboring the top 2 most abundant genes, *gcpE* and *dps*,

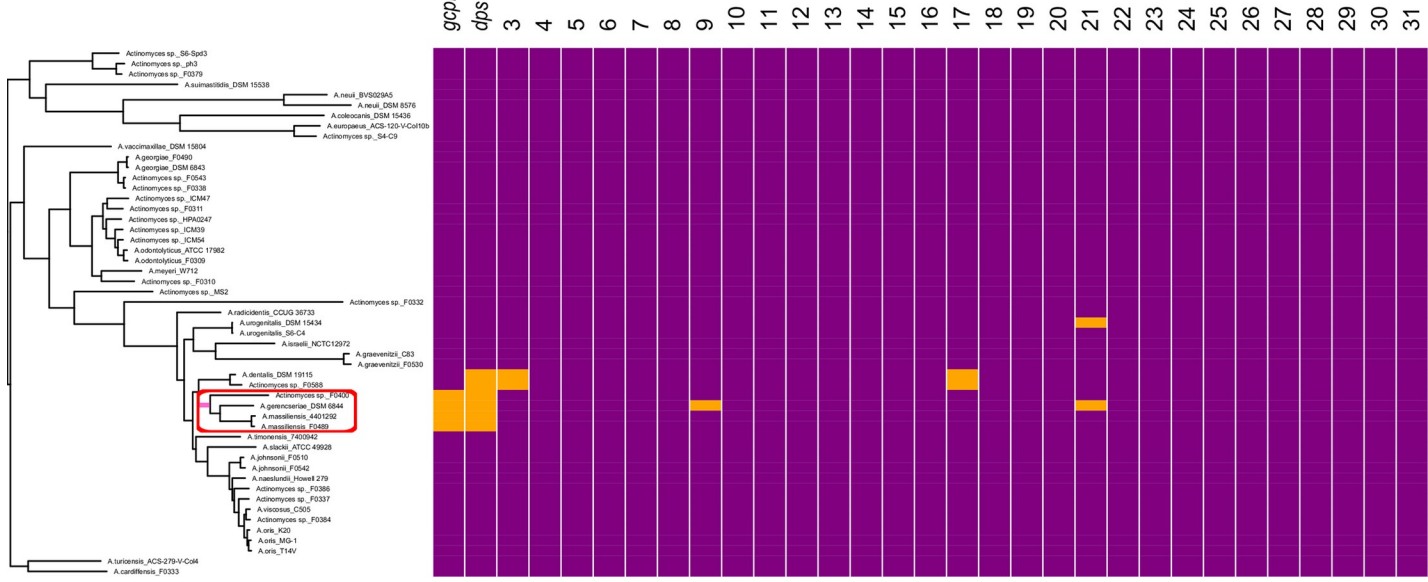

**Fig 3. Phylogenetic distribution of the MRONJ-associated genes in the core genome phylogeny of *Actinomyces*.** Strains in the tree are listed in S2 Table. Each column indicates a MRONJ-associated gene ordered in terms of the abundance of the gene across the 10 MRONJ samples (Fig 2B). Gene names are indicated for the top 2 genes. In the heatmap at the right, orange and purple indicate presence and absence of the genes, respectively. The red circle indicates a notable sub-clade mentioned in the main text. Pink indicates a branch separating the sub-clade from the others with 100% bootstrap support.

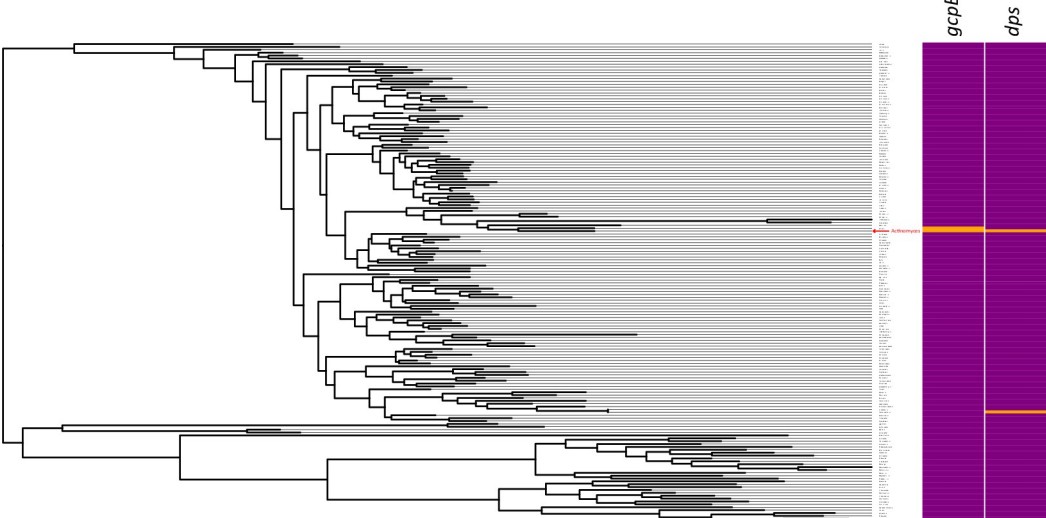

**Fig 4. Genus-level broad phylogenetic distribution of the 4 MRONJ-associated genes.** 16S rRNA phylogenetic tree including various genera. Strains in the tree are listed in **S3 Table**. In the heatmap at the right, the orange and purple indicate presence and absence of the genes, respectively.

were exclusively found in a sub-clade (red circle in Fig 3) with 100% bootstrap support (pink branch in Fig 3). The sub-clade included an *Actinomyces gerencseriae* strain and two *Actinomyces massiliensis* strains, as well as an *Actinomyces* strain that did not have a species name. The two species were confined to this sub-clade in the whole *Actinomyces* phylogeny.

To explore the phylogenetic distribution of the top 2 genes except *Actinomyces*, a BLASTn search was performed against the entire Kyoto Encyclopedia of Genes and Genomics (KEGG) database. The genes were all present only in genus *Actinomyces* (Fig 4, red arrow), as shown in the tree constructed using 16S rRNA gene sequences among various genera (Fig 4 and S3 Table). E-values of the BLASTn searches as well as abundance of each genus in oral microbiota of each sample calculated from the taxonomic profiling are shown in S3 Table. *gcpE* was detected only in *Actinomyces* and *Schaalia*, and *dps* only in *Actinomyces* and *Propionibacterium*. Compared to *Actinomyces* with on average 5.3% abundance across the 5 healthy control samples, *Schaalia* and *Propionibacterium* were not usual members of human oral microbiome because their abundances were always zero or < 0.5% (S3 Table).

## Discussion

*Actinomyces* was increased by an average of 10.1% in MRONJ patients when compared to healthy controls. The result was consistent with a previous study [14] indicating that *Actinomyces* is dominant in oral microbiome of MRONJ patients. The present approach was more quantitative due to the shotgun metagenomic sequencing. Gene-level analyses identified 31 MRONJ-associated genes. All were encoded by *Actinomyces* genomic regions. Among them, the top 2 most abundant genes were *gcpE* and *dps* that were all present only in the sub-clade of *Actinomyces*. Their hypothetical relationships with the disease at the molecular level based on literature data is depicted in Fig 5.

*gcpE* encodes an iron-sulfur enzyme, which synthesizes (E)-4-hydroxy-3-methylbutyl-2-enyl pyrophosphate (HMBPP). *GcpE* is inactivated by dioxygen [35]. It could be activated and decreased in the blood by using drugs via reduction of bone resorption or inhibition of angiogenesis [36]. The activation of *gcpE* increases the synthesis of HMBPP, which is a potent

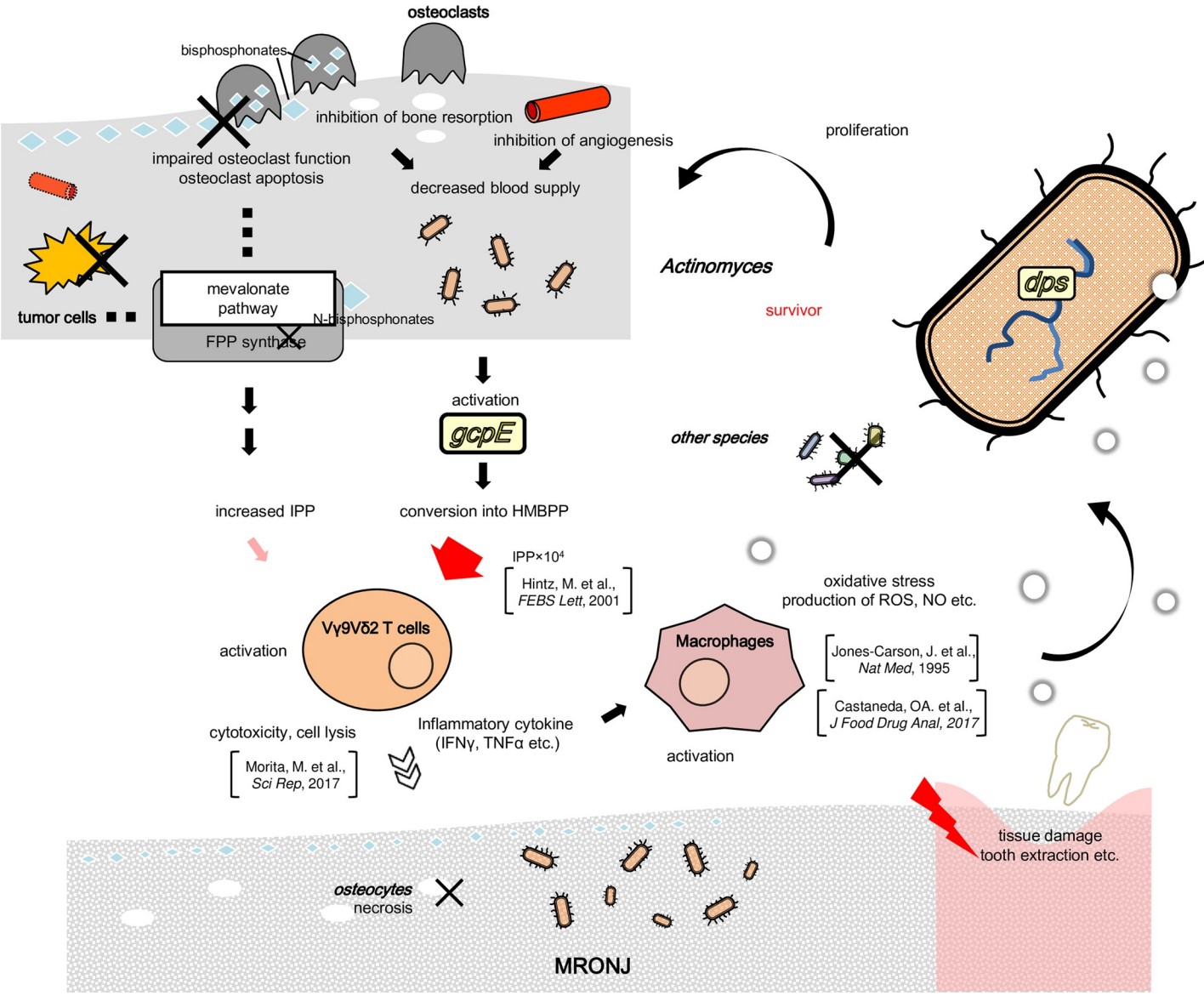

**Fig 5. Schematic representation of potential relationships between the 4 MRONJ-associated genes and the disease at the molecular level.** The 4 genes are indicated as yellow circles. Abbreviations: FPP: farnesyl diphosphate, HMBPP: (E)-4-hydroxy-3-methylbutyl-2-enyl pyrophosphate, IPP: isopentenyl diphosphate, NO: nitric oxide, and ROS: reactive oxygen species. Bisphosphonates and other drugs could trigger osteosclerosis or block angiogenesis. *GcpE* synthesizes HMBPP and is usually functionally inactivated by dioxygen. The gene could be activated under the condition of the lack of blood supply. HMBPP activates Vγ9Vδ2 T cells. The effect is $10^4$ greater than IPP that is accumulates when bisphosphonates are used. Vγ9Vδ2 T cells have strong cell lytic activity to induce osteocyte apoptosis and secretion of cytotoxic substances, which can lead to MRONJ. Activated cells also produce inflammatory cytokines, such as tumor necrosis factor-alpha (TNF-α) and interferon-gamma (IFN-γ). This strong local inflammation can activate macrophages, which produce reactive oxygen species (ROS) and nitric oxide (NO). ROS and NO are capable of damaging DNA, proteins, and other cellular compartments, and can impose potentially lethal stress on bacteria. *Actinomyces* can survive and proliferate, whereas the other bacteria have no defenses and decrease in frequency.

activator of human Vγ9Vδ2 T cells (a major subpopulation of γδ T cells). The effect of HMBPP on human Vγ9Vδ2 T cells stimulation is $10^4$-fold greater (large red arrow in Fig 5 than isopentenyl diphosphate (IPP) [37] that accumulates when nitrogen-containing BPs are used ("increased IPP" to the left of *gcpE* in Fig 5). Vγ9Vδ2 T cells have pronounced cell lytic activity. This induces target cell apoptosis and secretion of cytotoxic substances, which are active against both the infected host cells as well as pathogens [38]. Activated γδ T cells also

produce inflammatory cytokines, such as tumor necrosis factor-alpha and interferon-gamma that are detected in MRONJ lesions compared to control in vitro [39]. This strong local inflammation could contribute to osteonecrosis in MRONJ [40], consistent with a previous study that reported osteonecrosis in infectious osteomyelitis in mice [40].

*dps* encodes a dodecameric (12-mer) bacterial ferritin that protects DNA from oxidative stress and has been implicated in bacterial survival and virulence [41]. The γδ T cells activated by HMBPP and IPP produce inflammatory cytokines, which activate macrophages that produce reactive oxygen species (ROS) through NADPH oxidase and nitric oxide (NO) through inducible nitric synthase (iNOS) in response to bacterial infection [42, 43]. ROS and NO are capable of damaging DNA, proteins, and other cellular compartments including those of bacteria. iNOS is also up-regulated by activation of macrophages by ɣδ T cells. iNOS is also involved in the process of wound healing [44], likely including surgical invasion, such as extraction of teeth.

In summary, by conducting taxonomic profiling and gene-level analyses in the oral microbiota, we identified the 2 key genes almost specific to MRONJ and encoded by *Actinomyces*. Because the sample size used in this study was small, these preliminary data require further verification. The findings of this study can, however, aid future studies to verify the data and understand the disease mechanisms, develop molecular targeted drugs, and be useful for early stage screening and prognosis prediction.

## Supporting information

**S1 Table. The 17 MRONJ-associated genes.**
(XLSX)

**S2 Table. *Actinomyce*s strains used for construction of core-genome phylogeny and presence or absence of the 17 MRONJ-associated genes in each strain.**
(XLSX)

**S3 Table. Bacterial genera carrying at least one of the top 2 most abundant MRONJ-associated genes.**
(XLSX)

## Acknowledgments

The computational calculations were done at the Human Genome Center at the Institute of Medical Science (University of Tokyo) and at the National Institute of Genetics. We are grateful to Editage (www.editage.jp) for English language editing.

## Author Contributions

**Conceptualization:** Hiroko Yahara.

**Data curation:** Hiroko Yahara, Akimitsu Hiraki, Yutaka Maruoka, Aki Hirabayashi, Masato Suzuki, Koji Yahara.

**Formal analysis:** Hiroko Yahara, Koji Yahara.

**Funding acquisition:** Hiroko Yahara, Koji Yahara.

**Investigation:** Hiroko Yahara, Koji Yahara.

**Methodology:** Hiroko Yahara, Koji Yahara.

**Project administration:** Hiroko Yahara, Koji Yahara.

**Resources:** Akimitsu Hiraki, Yutaka Maruoka, Aki Hirabayashi, Masato Suzuki.

**Software:** Hiroko Yahara, Koji Yahara.

**Supervision:** Hiroko Yahara.

**Validation:** Hiroko Yahara, Koji Yahara.

**Visualization:** Hiroko Yahara, Koji Yahara.

**Writing – original draft:** Hiroko Yahara, Koji Yahara.

**Writing – review & editing:** Hiroko Yahara, Koji Yahara.

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
