## [Decision Letter · Decision Letter 0]

14 Sep 2020

PONE-D-20-26471

Shotgun metagenome sequencing identification of a set of genes encoded by Actinomyces specific to medication-related osteonecrosis of the jaw

PLOS ONE

Dear Dr. Yahara,

Thank you for submitting your manuscript to PLOS ONE. After careful consideration, we feel that it has merit but does not fully meet PLOS ONE’s publication criteria as it currently stands. Therefore, we invite you to submit a revised version of the manuscript that addresses the points raised during the review process.

Both reviewers have positive attitudes to your study. Reviewer #2, however, has pointed out that the statistical reliability of the results is minimally OK and that additional samples should be analyzed.  I, not expert in the field can imagine that additional samples could be obtained  and analyzed to test the frequency of the correlation.

We look forward to receiving your revised manuscript.

Kind regards,

Ulrich Melcher

Academic Editor

PLOS ONE

Journal Requirements:

2.Thank you for including the following ethics statement on the submission details page:

'This study was approved by the ethics committees of the Research Institute National

Center for Global Health and Medicine (approval number NCGM-G-0002529-02),

National Institute of Infectious Diseases (811), Fukuoka Dental University (368).

Written informed consent was obtained.'

Please also include consent information in the ethics statement in the Methods section of your manuscript.

3.We note that you have indicated that data from this study are available upon request. PLOS only allows data to be available upon request if there are legal or ethical restrictions on sharing data publicly. For more information on unacceptable data access restrictions, please see http://journals.plos.org/plosone/s/data-availability#loc-unacceptable-data-access-restrictions.

4.PLOS requires an ORCID iD for the corresponding author in Editorial Manager on papers submitted after December 6th, 2016. Please ensure that you have an ORCID iD and that it is validated in Editorial Manager. To do this, go to ‘Update my Information’ (in the upper left-hand corner of the main menu), and click on the Fetch/Validate link next to the ORCID field. This will take you to the ORCID site and allow you to create a new iD or authenticate a pre-existing iD in Editorial Manager. Please see the following video for instructions on linking an ORCID iD to your Editorial Manager account: https://www.youtube.com/watch?v=_xcclfuvtxQ

<h1>** **</h1>

Reviewers' comments:

Reviewer's Responses to Questions

**Comments to the Author**

1. Is the manuscript technically sound, and do the data support the conclusions?

Reviewer #1: Yes

Reviewer #2: Partly

2. Has the statistical analysis been performed appropriately and rigorously? 

Reviewer #1: Yes

Reviewer #2: N/A

3. Have the authors made all data underlying the findings in their manuscript fully available?

Reviewer #1: Yes

Reviewer #2: Yes

4. Is the manuscript presented in an intelligible fashion and written in standard English?

Reviewer #1: Yes

Reviewer #2: Yes

5. Review Comments to the Author

Reviewer #1: This is a really well realized paper.

This paper analyzes an interesting and modern aspect of the pathogenesis of MRONJ.

The presence of actinomyces is a frequent finding in BRONJ.

Previously it was believed to be present only in osteonecrosis related to the use of bisphosphonates (BRONJ), recently it has also been detected in osteonecrosis related to denusumab (MRONJ), (as in the series that my group has just published Cerrato A, Zanette G, Boccuto M, Angelini A , Valente M, Bacci C. Actinomyces and MRONJ: A retrospective study and a literature review [published online ahead of print, 2020 Aug 20]. J Stomatol Oral Maxillofac Surg. 2020; S2468-7855 (20) 30175-0. Doi: 10.1016 / j.jormas.2020.07.012 which the Authors can quote if they want).

The present paper further supports the infectious theory of MRONJ bringing an important contribution to the knowledge of the pathology.

Reviewer #2: The manuscript by Yahara et al focusses on potential bacterial causes of MRONJ by deep shotgun metagenome sequencing of saliva samples. With the method they used, they could overcome the limitations in previous studies, such as unculturable taxa in human oral microbiome or lacking the quantitative information on abundance of specific taxa. They analyzed saliva samples collected from 5 MRONJ patients and 4 healthy volunteers, and revealed that the genus Actinomyces was 10.1% more abundant in MRONJ patients than in healthy volunteers. They also determined the 119 bacterial genes specifically found in these patients, and showed that all of them were encoded by Actinomyces genus. Furthermore they also described the most abundant eight Actinomyces genes in MRONJ patients.

In general, the data presented in this study is interesting and would shed light on an unknown field which is the molecular connection between Actinomyces and MRONJ. Nevertheless, the manuscript needs a major revision.

Major issues

• A clear difference in bacterial composition between the samples from MRONJ patients and healthy donors could be demonstrated; however, the cohort is too small to draw any generalizable conclusions. I would suggest that at least a total of 20 samples per group should be evaluated. By taking the fact in account that MRONJ is a rare disease, the cohort can be enlarged by establishing collaborations with other centers. Still, the authors should discuss the limitations of the study, particularly because of small sample number and present their data as a pilot study in this area which needs to be confirmed further.

• Having the reads from only two MRONJ patients presented in figure S2 compromises the reliability. The authors should collect the same data from all MRONJ patients and then recreate the figure for reliable interpretation. When all these requisites have been fulfilled, Figure S2 may be one of the main figures since determination of Actinomyces-specific genes found predominantly in MRONJ patients is an important research question in this study.

• The manuscript is well-written and easy to follow except from the discussion part. This section should be more compactly structured to lead the reader to follow the story depicted in Figure 4.

 

Minor issues

• It would be easier to follow if the figure legends were presented with the figures in the manuscript.

• There’re typos in Figure S2 axis labeling.

• Lines 278-280: …orange and purple indicate presence and absence of the genes. Missing “, respectively”?

• Line 285: “except” instead of “outside”

6. PLOS authors have the option to publish the peer review history of their article (what does this mean?). If published, this will include your full peer review and any attached files.

Reviewer #1: **Yes: **Christian Bacci

Reviewer #2: No

---

## [Author Response · Author response to Decision Letter 0]

13 Oct 2020

Uploaded as a Word file and included in the merged PDF

---

## [Editor Report · Decision Letter 1]

20 Oct 2020

Shotgun metagenome sequencing identification of a set of genes encoded by Actinomyces associated with medication-related osteonecrosis of the jaw

PONE-D-20-26471R1

Dear Dr. Yahara,

We’re pleased to inform you that your manuscript has been judged scientifically suitable for publication and will be formally accepted for publication once it meets all outstanding technical requirements. The comments of the previous reviews have been more than adequately addressed. However: line 349 should be”its’ rather than “is.

Kind regards,

Ulrich Melcher

Academic Editor

PLOS ONE
---

## [Editor Report · Acceptance letter]

13 Nov 2020

PONE-D-20-26471R1 

Shotgun metagenome sequencing identification of a set of genes encoded by *Actinomyces* associated with medication-related osteonecrosis of the jaw 

Dear Dr. Yahara:

I'm pleased to inform you that your manuscript has been deemed suitable for publication in PLOS ONE. Congratulations! Your manuscript is now with our production department. 

Kind regards, 

on behalf of

Dr. Ulrich Melcher 

Academic Editor

PLOS ONE